# Transcriptomic Response of the Diazotrophic Bacteria *Gluconacetobacter diazotrophicus* Strain PAL5 to Iron Limitation and Characterization of the *fur* Regulatory Network

**DOI:** 10.3390/ijms23158533

**Published:** 2022-08-01

**Authors:** Cleiton de Paula Soares, Michelle Zibetti Trada-Sfeir, Leonardo Araújo Terra, Jéssica de Paula Ferreira, Carlos Magno Dos-Santos, Izamara Gesiele Bezerra de Oliveira, Jean Luiz Simões Araújo, Carlos Henrique Salvino Gadelha Meneses, Emanuel Maltempi de Souza, José Ivo Baldani, Marcia Soares Vidal

**Affiliations:** 1Embrapa Agrobiologia, Rodovia BR 465, Km 07, Seropédica 23891-000, RJ, Brazil; cleiton_depaula@yahoo.com.br (C.d.P.S.); leonardoterra@hotmail.com.br (L.A.T.); jeessica_ufrrj@yahoo.com.br (J.d.P.F.); jean.araujo@embrapa.br (J.L.S.A.); ivo.baldani@embrapa.br (J.I.B.); 2Departamento de Bioquímica e Biologia Molecular, Setor de Ciências Biológicas, Centro Politécnico, Universidade Federal do Paraná, Jardim das Américas, Curitiba 81531-980, PR, Brazil; miztadra@yahoo.com.br (M.Z.T.-S.); souzaem@ufpr.br (E.M.d.S.); 3Programa de Pós-Graduação em Ciências Agrárias, Departamento de Biologia, Centro de Ciências Biológicas e da Saúde, Universidade Estadual da Paraíba, Bairro Universitário, Campina Grande 58429-500, PB, Brazil; c.magno.s@hotmail.com (C.M.D.-S.); izamara.oliveira@aluno.uepb.edu.br (I.G.B.d.O.); carlos.meneses@gsuite.uepb.edu.br (C.H.S.G.M.)

**Keywords:** endophytic bacteria, plant growth-promoting bacteria, mRNA-Seq, *fur* regulon, iron homeostasis

## Abstract

*Gluconacetobacter diazotrophicus* has been the focus of several studies aiming to understand the mechanisms behind this endophytic diazotrophic bacterium. The present study is the first global analysis of the early transcriptional response of exponentially growing *G. diazotrophicus* to iron, an essential cofactor for many enzymes involved in various metabolic pathways. RNA-seq, targeted gene mutagenesis and computational motif discovery tools were used to define the *G. diazotrophicus*
*fur* regulon. The data analysis showed that genes encoding functions related to iron homeostasis were significantly upregulated in response to iron limitations. Certain genes involved in secondary metabolism were overexpressed under iron-limited conditions. In contrast, it was observed that the expression of genes involved in Fe-S cluster biosynthesis, flagellar biosynthesis and type IV secretion systems were downregulated in an iron-depleted culture medium. Our results support a model that controls transcription in *G. diazotrophicus* by *fur* function. The *G. diazotrophicus*
*fur* protein was able to complement an *E. coli*
*fur* mutant. These results provide new insights into the effects of iron on the metabolism of *G. diazotrophicus*, as well as demonstrate the essentiality of this micronutrient for the main characteristics of plant growth promotion by *G. diazotrophicus*.

## 1. Introduction

Iron is an essential micronutrient for most microorganisms. However, it presents problems such as toxicity and solubility [1]. Iron bioavailability is low in cultivated soils, in the rhizosphere, as well as in plants; therefore, there is strong competition among microorganisms to fill these niches [2]. This evidence points toward iron as an important element in bacterial attachment, biofilm formation, competitive rhizosphere colonization, and colonization of host plant tissues [3,4,5,6].

*Gluconacetobacter diazotrophicus* species is a Gram-negative bacterium belonging to the α-proteobacteria, and its main feature is its capacity for biological nitrogen fixation (BNF). This bacterium was originally isolated from the stems and roots of different varieties of sugarcane [7]. In addition, its ability to colonize *Arabidopsis thaliana*, sorghum, sweet potato, pineapple, coffee, elephant grass and rice has been reported [8,9,10,11,12,13,14]. 

The benefits of *G. diazotrophicus* in promoting sugarcane growth opened the possibility of using this diazotrophic bacteria as a sugarcane inoculant. A new perspective on the study and exploration of this biotechnological potential is the result of the genome publication of this bacterium [15], which allowed a better understanding of the role played by this bacterium in the plant–bacteria interaction.

*G. diazotrophicus* iron homeostasis study is of interest for several reasons. First, iron is an important nutrient, serving as a cofactor for proteins involved in respiration, the tricarboxylic acid (TCA) cycle, enzyme catalysis, gene regulation, BNF and DNA biosynthesis, in addition to being essential for host tissue colonization [3,5,16]. Iron not only acts as a cofactor in *G. diazotrophicus*, but also as an important element for growth, nitrogenase activity and biofilm formation [17,18].

In many bacteria, the Ferric-uptake regulator (*fur*) is involved in the coordinated regulation of gene expression in response to iron availability. Fur utilizes Fe^2+^ as a cofactor, represses the expression of iron uptake and metabolism genes under iron sufficiency and causes downregulation in the absence of Fe^2+^ under conditions of iron restriction. Fur-Fe^2+^ also has been reported to be involved in the positive regulation of expression of genes involved in iron storage proteins, superoxide dismutase, and catalase [19]. 

In addition, ferric iron binding transcription factor XibR, which belongs to the NtrC family of proteins in *Xanthomonas campestris*, plays a dual role in iron metabolism, suppresses siderophore expression under iron-replete conditions, and positively regulates the expression of outer membrane receptors for iron/iron complex uptake [20]. Like *fur* and DtxR, IdeR binds iron and then interacts with a specific sequence in the operator regions of iron-regulated genes to control their transcription. Generally, IdeR acts as a global regulator coordinating iron homeostasis, morphological differentiation, central metabolism, secondary metabolism, and oxidative stress responses [21].

We have previously demonstrated, using a *G. diazotrophicus* mutant for the *tonB* gene, that the TonB complex of *G. diazotrophicus* has a role in the transport of Fe^3+^-siderophore complexes and may have an essential function in the endophytic lifestyle of this organism [18]. The aim of the present work was to determine the transcriptional response of *G. diazotrophicus* to iron availability and to investigate the role of the ferric uptake regulator Fur in *G. diazotrophicus* iron homeostasis. For this purpose, we applied RNA-seq, RT-qPCR, *lacZ* reporter fusions, *fur* insertional mutants and computer-aided genome searches to characterize the *G. diazotrophicus fur* regulon. The results of this work increased our understanding of the responses to iron deficiency in *G. diazotrophicus*.

## 2. Results

### 2.1. Differentially Expressed Gene Identification and Iron-Activated and Repressed Genes by Functional Category

To further advance our knowledge related to iron-regulated genes in the *G. diazotrophicus* strain PAL5, the changes induced in the transcriptome of iron-starved PAL5 following iron treatment were investigated. The transcriptional profile of PAL5 in the presence and absence of iron generated an average of approximately 14 million reads for both treatments, with approximately 2 million mapped only to the genome of PAL5 (NCBI accession number NC_010125.1, NC_010124.1 and NC_010123.1) when cultured in iron-starved PAL5 cells induced by iron (Appendix A).

The biological replicates showed a remarkably high correlation level within each treatment with regard to the ORF-mapped read counting (r^2^ > 0.99) (Appendix A). In the transcriptomic analysis, 324 genes were identified, representing 9.28% of the PAL5 genome. These included 96 (2.75%) upregulated and 228 (6.53%) downregulated genes, at least a 1.2-fold change, after exposure to iron-limiting conditions (Appendix A).

In total, 56 Gene Ontology (GO) terms were assigned to 324 genes. Most of the GO terms were attributed to molecular function (39.3%), followed by biological process (35.7%) and cell sharing (25%). The terms GO were derived from 30 different functional groups (GO subcategories, level 3) (Figure 1). The largest subgroups within the biological process superclass were assigned to “cellular metabolic process,” “organic substance metabolic process,” “primary metabolic process” and “nitrogen compound metabolic process,” while within the molecular function superclass were “heterocyclic compound binding,” “organic cyclic compound binding,” “transferase activity,” “oxidoreductase activity,” and “ion binding.” The majority within the cellular component superclass were assigned the categories “intrinsic component of membrane”, “intracellular” and “intracellular part”.

Regarding COG, most upregulated genes belong to the functional classes of function unknown, carbohydrate transport and metabolism, energy production and conversion. (Figure 2). On the other hand, downregulated genes belong to the cell wall/membrane/envelope biogenesis, transcription, general function prediction only, and function unknown. (Figure 2). A large group of genes belonging to the category of not detected have been identified.

### 2.2. Main Metabolic Changes in Bacterial Adaptation to Iron-Limiting Conditions

Many genes undergoing transcriptional changes were known or predicted to be involved in iron homeostasis (Figure 3). When comparing gene expression between LGI-P media without or with added iron, the results showed that iron stimulus controls genes that are important for metabolic activities in PAL5 (Appendix A).

Three genes coding for TonB-dependent siderophore receptors were differentially expressed, the gene *fiu,* a catecholized siderophore receptor, was upregulated and two copies of the *CirA* (GDI_3807 and GDI_1948) were downregulated in the transcriptome analysis. Cultivation in the presence of iron also upregulated genes encoding biopolymer transport proteins (*exbB* and *tolC*), *omp* (GDI_0807), *fpr* (GDI_2217) that codes for ferredoxin reductase, and one ortholog of the NRAMP family transporters known in bacteria as *mntH* (GDI_0654). It also downregulated genes encoding bacterioferritin (GDI_3449), *Fdx* (2Fe-2S Ferredoxin), *CoxS* [(2Fe-2S)-binding protein] and *HybA* (4Fe-4S ferredoxin). 

The results for the major facilitator superfamily (MFS) transporter include the *tppB* gene, which encodes putative tripeptide permease and GDI_0037 that was downregulated. However, only the *galP* gene that encodes for a galactose-proton symporter was overexpressed. In addition, several genes coding for efflux pumps of resistance-nodulation-division (RND) multidrugs were upregulated. These include the operon genes for *mdtABC* and *acrAB,* which were downregulated and overexpressed, respectively.

Among the overexpressed ATP-binding cassette (ABC) transporter genes, there were three for saccharide, polyol and lipid transporters (*ugpA*, *rbsA* and *rbsC*), two for phosphate and amino acid transporters (*pstB* and *pstC*) and one peptide and one nickel transporter (GDI_0466). Among the downregulated transporters, there were one cell division transporter (GDI_3296), one for amino acid transporters (GDI_3296), and one for oligopeptide and nickel transport systems (GDI_0248).

The *trbB* (GDI_2745) and *trbD* (GDI_2912) genes involved in assembling the filament of type IV secretion systems (T4SS) were suppressed after iron treatment. Curiously, members of the peroxiredoxin family of antioxidant peroxidases encoded in the *G. diazotrophicus* genome, *bcp* (GDI_2772), *ahpD* (GDI_0576) alkyl hydroperoxide reductase and *ahpF* (GDI_0773) alkyl hydroperoxide reductase subunit F, were downregulated under this condition, as were transcripts for *cspA* (GDI_3810), which codes for a cold heat shock protein. In contrast, transcripts for thioredoxin, thioredoxin reductase (GDI_3513), *GroS*, *gsT*, *xdhC* and *degQ* gene were upregulated in the presence of iron. RNA polymerase for a specialized set of genes in response to ecophysiological signals, such as *rpoH*, was also upregulated. 

Genes encompassing energy generation encoding pyruvate dehydrogenase E1 alpha and beta subunits (GDI_1939, GDI_1940) and phosphoenolpyruvate dikinase (GDI_1318) were upregulated in the presence of iron. The *sucCD* genes (GDI_2951; GDI_2952) involved in TCA, proponoate and C-5 Branched were also upregulated. In addition, the *cyoA* gene (GDI_2035), which codes for cytochrome ubiquinol oxidase subunit II, and *cyoB* (GDI_2101), which codes for cytochrome-c oxidase, showed positive regulation. However, the *nuoI* NADH-quinone oxidoreductase subunit I (GDI_3032), *nuoK* NADH-quinone oxidoreductase subunit B (GDI_3310) and *atpH* ATP synthase subunit delta (GDI_0693) involved in oxidative phosphorylation were downregulated.

Altogether, 10 related cell wall genes were downregulated. Among the downregulated genes, two copies of the *rfaG* genes coding for glycosyltransferases were found, as well as two copies of *mltE* genes coding for lytic transglycosylase, *lpxK a*-tetraacyldisaccharide 4’-kinase, *mrcA* a penicillin-binding protein 1A, *asmA*, *wcaG* an epimerase and *ompA* a peptidoglycan-associated lipoprotein. A gene encoding a Lipid A core-O-antigen ligase (*rfaL*) was upregulated.

Genes involved in bacterial chemotaxis and flagellar structure *fliQ* (GDI_1658), *flhB* (GDI_1660), *fliG* (GDI_1677) and *fliP* (GDI_1681) were widely downregulated in the strain PAL5 cultured in absence of iron. In addition, under the tested conditions, the type I pilus components *cpaA*, *cpaF* and GDI_1704 (putative *Flp*/*Fap* pilin component) were downregulated. Interestingly, only one *cheD* gene, a putative chemotaxis protein, *cheY*, was upregulated.

Genes involved in regulation, clustering Fe-S and BNF were differentially expressed. The *ntrX* (GDI_2263), *nrfA* (GDI_2262), *HesB* and *fxdB* genes (GDI_2370) were all downregulated, as were the *nifH* (GDI_0436) and *nifK* (GDI_0438) genes.

Seven transposase genes belonging to five different families (IS5/IS1182, IS21, IS110, and IS256) were upregulated. Transcriptional regulators were modulated, where 19 genes (6 *LysR*, 2 *MarR*, *CopG*, *HxlR*, *AsnC*, *GntR*, *MerR*, *Arac*, *NmrA*) were differentially expressed, 14 of which were downregulated in the presence of iron.

The data obtained by RNA-seq were validated by RT-qPCR using random genes and differentially expressed (*wrbA*, GDI_1972, GDI_0466, *fdx*, *groS*, *pstC*) when cultured in iron-limiting conditions. The reference genes *rhO* and *rpoD* were used as internal controls (Figure 4). 

### 2.3. Identification of fur Box Sequences in the G. diazotrophicus Genome

To identify genes directly regulated by *fur*, different weight matrices were used to locate potential *fur*-binding sites in the *G. diazotrophicus* genome. Several potential *fur*-binding sites were detected (Appendix A). Candidate *G. diazotrophicus fur*-binding sites derived from both bioinformatic prediction routines and RNA-seq were aligned and used to derive the corresponding logos (Figure 5). To obtain a more realistic estimate of the total number of genes regulated by *fur*, the genes associated with *fur* binding sites were analyzed regarding their presence in the transcriptome in response to iron availability (Figure 6). 

Predicted *fur*-binding sites were identified upstream of several genes and/or operons encoding proteins involved in iron acquisition and transportation (GDI_3807, GDI_0553, GDI_0207 GDI_1223, *tonB*, *feuA*, *feuB*), NADH dehydrogenase (*nuoA*, *nuoB*, *nuoCD*, *nuoE*, *nuoF*, *nuoG, nuoH*, *nuoI*, *nuoJ*, *nuoK*, *nuoL*, *nuoM*, *nuoN*, *lao*), succinate dehydrogenase (*sdh*), aconitate hydratase (*acnA*), cytochrome c oxidase (*cyoA*, *cyoB*), tryptophan biosynthesis (*tprB*, *trpA*, *aldA*) and enzyme nitrogenase (*nifA*, *nifB*, *nifD*, *nifH*) and *LysR* transcriptional regulator (GDI_0187).

The DNA sequence logos derived from all predicted *fur*-binding sites are represented in Figure 4. The consensus sequence contains a 19-bp palindromic motif (9-1-9 inverted repeat) that is highly similar to the previously described *fur*-box [22].

### 2.4. Production of Siderophores in Response to Iron Limitations

According to the data in Figure 7, *G. diazotrophicus* PAL5 showed an increase in the production of siderophores when cultivated in the LGI-P medium without the presence of iron (47% siderophores). On the other hand, the level of siderophores drastically reduced in the presence of 37 µM FeCl3 (16 % siderophores).

### 2.5. fur Insertional Mutants and Complementation of an E. coli fur Mutant by G. diazotrophicus fur Homologs

To address the physiological role played by *fur* in *G. diazotrophicus*, attempts to generate *fur*-null bacterial mutants were made but unsuccessful, suggesting that mutation in the *fur* gene (single copy) is lethal to the PAL5 strain. However, complementation of *E. coli* H1780 with pGGdfur (from strain PAL5) rescued the *fur* defect of this strain and resulted in repression of the transcribed *fiu-lacZ* reporter gene (Figure 8). 

Since the complementation of the *E. coli* H1780 *fur* mutant strain is related to the transcription of the *fiu*-*lacZ* reporter gene, the β-Galactosidase activity was measured accurately (Figure 9). The results showed that the β-Galactosidase activity is much higher in constructs that lack functional Fur protein *E. coli* H1780 and *E. coli* H1780 (pGGdfurTn5). The *E. coli* H1780 (pGGdfur), the functional Fur protein, when in the presence of iron, forms the Fur-Fe^2+^ complex and blocks the transcription of the *fiu*-*lacZ* gene and the enzyme is not encoded. In the absence of iron, due to the addition of 2,2′-dipyridyl, an iron-chelating agent, the enzyme activity was higher, as there was no iron available in the medium, and Fur-Fe^2+^ complex transcription was not blocked.

## 3. Discussion

### 3.1. Iron Availability Promotes Changes in the G. diazotrophicus Transcriptome

Iron homeostasis is extremely important to bacteria and is believed to be required to enable plant growth promotion; however, the existing information on iron homeostasis is insufficient to understand this phenomenon. de Paula Soares et al. [18] showed that iron homeostasis plays an essential function in the endophytic lifestyle.

First, to define the iron *G. diazotrophicus* limitation stimulus, we compared the growth curve of PAL5 wild-type cells treated with 37 μM FeCl_3_ with that of wild-type cells treated with 0 μM FeCl_3_ for 48 h. Our results from the differential expression experiments showed that most of these genes related to iron homeostasis in PAL5 were completely repressed (227 genes) and the other induced partially (96 genes) under iron-limiting conditions, reflecting the rigid regulation of these genes (Appendix A). This pattern of gene expression in the first 30 min after adding iron, preceded by a long period of iron starvation, appears to reveal the modulation of essential genes that respond quickly to iron hunger and allow the expression of the right genes to restore iron homeostasis in the cell.

### 3.2. Metabolic Pathways and Iron Uptake

As expected, in our in vitro assays in which *G. diazotrophicus* grew under iron-limiting conditions, catecholate siderophore-mediated acquisition systems (*fiu*) and TonB-dependent receptors (e.g., *cirA*) appeared to be the most successful strategies for overcoming iron limitations. Similar expression profiles of *fiu* and *cirA* were observed in *Herbaspirillum seropedicae* under iron-limiting conditions [23]. It is important to highlight the induction of the *exbB* gene that is part of the *tonB-exbB-exbD* operon; the TonB–ExbB–ExbD protein complex participates in the active transport of iron–siderophore complexes, nickel complex, carbohydrates and vitamin B12 across the outer membrane [24]. The transcription of *tonB* and *exbD* genes in *G. diazotrophicus* was also upregulated upon iron limitation but missed the cut-off criteria. In previous experiments, we used RT-qPCR to measure *tonB-exB-exbD* gene transcription in iron-limited conditions and provided evidence that the TonB complex of *G. diazotrophicus* plays a role in the transport of Fe^3+^-siderophore complexes [18].

In addition to *TonB*-*ExbB*-*ExbD*, some bacteria have an analogous system called *TolA*-*TolQ*-*TolB* [25]. The genes that code for the *tolQ* and *tolB* proteins are structurally and functionally similar to *exbB* and *exbD*, respectively, and were upregulated in the presence of iron. It is noteworthy that the *tolQ* gene had a high number of transcripts in the transcriptomic analysis. Interestingly, the *omp* gene, which is part of the same operon as the *tolB* gene, was highly expressed in both analyses, with a greater number of transcripts in the absence of iron. *TolA*-*TolQ*-*TolR*, in addition to providing necessary energy for the transport of colicins and cobalamin in *E. coli*, may also be involved in the absorption of sugars, polyols and amino acids [26].

The adaptive response of PAL5 to iron starvation included the under expression of defense systems genes. Among the genes repressed by iron limitation are multidrug resistance (MDR) efflux pumps. Our hypothesis is that the repression of these genes is part of the bacterial response to the negative energy balance caused by the long period of unavailable iron. The activity of an efflux pump depends on the different types of energy sources that each MDR system uses [27]. 

Most of the TonB-dependent transporters in the PAL5 strain, however, do not appear to be involved in the assimilation of iron, but rather in the intake of sugars, lipids, peptides, proteins and drugs across the cytoplasmic membrane. Apparently, there was a balance in the activation or repression of genes related to carbon and transport systems in PAL5 cells. However, a larger number of ABC-type sugar transporters were upregulated after the addition of iron. A sugar ABC transporter periplasmic protein was upregulated in *H. seropedicae* under iron starvation conditions [23]. This balance could be an artifact of the bacteria saving energy, since their metabolism is very active and the initial iron deficiency, an enzymatic cofactor for many central carbon metabolic reactions, affects the energetic balance, resulting in low activity of carbon metabolism.

This study demonstrated that the *bfd* gene that codes for bacterioferrin associated with iron homeostasis (2Fe-2S) (GDI_3449) was expressed more in the absence of iron. In contrast, the *fpr* gene (GDI_2217) that codes for ferredoxin reductase was expressed more in the presence of iron. Similar results were observed for *Pseudomonas aeruginosa* [28,29]. The Bfd accepts the electrons from the Fpr and transfers them to the Fe^3+^ which is stored in the heme-containing bacterioferrins, allowing the release and mobilizing Fe^2+^ to the cytosol [30,31].

Iron has chemical properties that can pose challenges, such as the generation of the hydroxyl radical [32]. Our data suggest that *G. diazotrophicus* does not suffer oxidative stress after 30 minutes of iron being made available, since no induction of superoxide dismutase (*sodA*) or catalase type E (*katE*) was observed. In addition, genes with well-known roles in defense against oxidative stress were also downregulated, including *bcp* (GDI_2772), *ahpD* (GDI_0576) alkyl hydroperoxide reductase and *ahpF* (GDI_0773) alkyl hydroperoxide reductase subunit F. Other genes, such as *gsT*, were upregulated but with low levels of transcription. It is possible that the concentration of iron used in this study was not sufficient to induce oxidative stress or that *G. diazotrophicus* had not yet responded to oxidative stress during the period examined. 

Interestingly, heat shock genes were also upregulated by iron limitation, as were some genes encoding peptidases containing metals as cofactors, which is consistent with previous observations in *Shewanella oneidensis* [33], *Caulobacter crescentus* [34] and *Piscirickettsia salmonis* [35]. Induction of these genes might be directly mediated by the heat shock sigma factor RpoH (σ^32^), because their own *rpoH* gene is upregulated in iron limitation [34], which corroborates the induction of the *rpoH* gene observed in our results. In *G. diazotrophicus*, nothing is known about the transcriptional regulation of *rpoH*, but since *rpoH* expression is largely constitutive, it is likely that transcriptional regulation plays a significant role in response to stress that destabilizes and denatures proteins.

Most genes with positive iron effects appear to be involved in energy metabolism, such as subunits of respiratory chain complexes (e.g., *CyoA*, *CyoB*), enzymes in the TCA (e.g., *SucC*, *SucD*), and glycolysis (e.g., *AcoA*, *AcoB*). Many of these proteins are used as part of a prosthetic group, such as the Fe-S cluster proteins, cytochromes, NADH dehydrogenases and succinyl-CoA synthetase. The assimilation of iron and the induction of genes essential for energy metabolism within the first few minutes of iron becoming available probably reflect essential nutritional needs. In fact, it has previously been observed that iron-containing proteins are upregulated under the iron conditions available in *E. coli* [36], *Campylobacter jejuni* [37], *P. aeruginosa* [29] and *H. seropedicae* [23]. Unexpectedly, NADH-quinone oxidoreductases (*nuoL* and *nuoK*), encoding proteins involved in oxidative phosphorylation, were downregulated by iron limitation. Although the reason why these genes were suppressed by iron starvation is not yet clear, it is reasonable to assume that these NADH-quinone oxidoreductases are necessary and upregulated only in the final phase of the cell respiration process after the first stages of glycolysis and TCA.

A drastic alteration in the expression of genes involved in cell wall or cell membrane biogenesis was observed. Genes encoding glycosyltransferases (*rfaG*), penicillin-binding protein (*mrcA*), lytic transglycosylase (*mltE*) and peptidoglycan-associated lipoprotein (*ompA*) were downregulated. The synthesis of bacterial cell wall peptidoglycan requires glycosyltransferase enzymes to transfer the disaccharide-peptide from lipid II onto the growing glycan chain and later growth for bacterial [38]. Under similar conditions of iron restriction, *Riemerella anatipestifer* CH-1 [39] and *Clostridioides difficile* [40] also showed changes in the genes involved in cell wall biogenesis. On the other hand, *OmpA* production is stimulated under iron-rich conditions, which suggests its potential role in iron metabolism, as observed in *Acinetobacter baumannii* [41].

It is unclear exactly how *G. diazotrophicus* couples flagellar gene expression with iron levels; however, previous studies have shown that several genes (*flaA, flaB, fliD*) appear to be directly regulated by the ferric uptake regulator Fur protein [40,42]. When bound to Fe^2+^, this intracellular iron sensor is capable of binding to *fur* boxes at operator sequences, blocking the transcription of genes involved in many cellular processes besides iron uptake. According to Rouws et al. [43] the *flgA* gene is involved in flagella biosynthesis, motility and biofilm-forming capacity in *G. diazotrophicus*. Repression of motility and chemotaxis genes by iron has been noticed in *Sinorhizobium meliloti* [44], *Acinetobacter baumannii* [45], *Pseudomonas fluorescens Pf- 5* [46], *C. crescentus* [34] and *Aliivibrio salmonicida* [47].

Genes involved in regulation, clustering Fe-S and BNF were found in our analysis. General *NtrBC* nitrogen regulation systems are signal transduction systems that detect and regulate nitrogen status. The *ntrX* gene that codes for the sigma-54-dependent *Fis* family transcriptionally showed a high number of transcripts in the absence of iron. Interestingly, a regulator of *nifA* (*nrfA*) [48], showed a higher number of transcripts in the absence of iron. *HesB,* a protein that has the Fe-S_biosyn domain and appears to be involved in the biogenesis of the Fe-S cluster, is regulated in the absence of iron. The *fxdB* gene, which is important for nitrogenase activity, was highly upregulated in the absence of iron. The *nifH* and *nifK* genes involved in the formation of the enzyme complex nitrogenase were identified as downregulated after the initial 30 minutes of available iron. A close relationship between iron homeostasis and BNF has been reported for nitrogen-fixing *H. seropedicae* [49,50], *Cyanobacterium Anabaena* sp. [51] and *G. diazotrophicus* [18]. The high content of iron found in many key proteins involved in nitrogen metabolism [52], limits this iron-uptake process. Therefore, the existence of regulatory interactions between iron and nitrogen metabolisms, probably orchestrated by *fur* [51], may also be a mechanism used by *G. diazotrophicus* to modulate nitrogen fixation in response to iron availability. *fur* gene was upregulated upon iron limitation but missed the cut-off criteria. The Fur protein is a key regulator in iron metabolism and in a previous study, transcript levels of the *fur* gene were downregulated in the presence of iron in PAL5 [18].

T4SS related to conjugation machinery, which generally involves a single-step secretion system and the use of a pilus [53], were downregulated. T4SS is an important component of adaptation, allowing horizontal gene transfer. In this respect, the involvement of transposases in the mobility of DNA is a major factor in the evolution of prokaryotes. Interestingly, seven genes belonging to five families of transposases (IS5/IS1182, IS21, IS110 and IS256 families) were also identified in our analyses as downregulated. These results suggest that iron plays a major role in the adaptation and evolution of *G. diazotrophicus*.

Our expression data indicate that plasticity of transcriptional machinery in *G. diazotrophicus* is represented by the repertoire of regulatory families modulated by iron. More specifically, we found that the *LysR*, *MarR*, *CopG*, *HxlR*, *AsnC*, *GntR*, *MerR*, *Arac* and *NmrA* families are mostly downregulated. Transcriptional regulators of the MarR and LuxR families in *Xylella fastidiosa*, MarR and GntR in *H. seropedicae* are modulated by iron [23,54]. However, some transcriptional regulators have been related to a wide diversity of functions. For example, the LysR-type transcriptional regulator (LTTR) family can be involved in metabolism, cell division, quorum sensing, virulence, motility, BNF, oxidative stress responses, toxin production, attachment and secretion, to name a few [55]. It is still unclear how totally different regulatory families play a very important role in the speciation of these organisms [56], nor how iron affects regulation. 

### 3.3. fur Regulatory Network

Transcriptomic results showed that 8.6% of the PAL5 genome is regulated in response to growth under iron-replete versus iron-depleted conditions. A similar result was obtained in *Neisseria gonorrhoeae* [57]. In silico analysis showed that 112 genes or operons were predicted to contain a *fur* box.

Among the genes within the *fur* box sequences identified in the regulatory region, some encode tonB-dependent receptors, such as *feuA* and *feuB*. In *G. diazotrophicus, feuABC* is organized in an operon and encodes proteins involved in the ABC-type iron transport system [17]. Also noteworthy is the presence of a *fur* box sequence in the regulatory region of the gene group *tonB-exbB-exbD1-exbD2,* between sites −35 and −10, more precisely at 300 bp of the codon initiator of the *tonB* gene [18]. Interestingly, we detected the presence of a *fur* box sequence upstream of a set of 15 genes probably organized in operon (*nuoA-nuoN* and *lao*), which constitute a putative NADH-ubiquinone oxidoreductase and L-amino acid oxidase, chain components of electron transport dependent on NADH, together with cytochrome C.

The occurrence of a *fur* box upstream of a series of *nif* genes suggests that *fur* acts in *G. diazotrophicus* as a regulator in the use of iron as a cofactor for the formation of the iron-protein and iron-molybdenum subunits of the enzyme nitrogenase. These results corroborate those observed in *Acidithiobacillus ferrooxidans* and *Shewanella oneidensis,* where *fur* positively regulates two important genes, *nifS,* an important cysteine desulfurase in the assembly of the Fe-S set and *nifX* a BNF protein important in the formation of the Fe-Mo subunit [58,59]. 

Our search for *fur* boxes in the genome of *G. diazotrophicus* yielded a substantial number of putative boxes that were differentially expressed by transcriptomic analyses. *fur* binding site prediction suggests that *fur* acts mainly as a direct transcriptional repressor in *G. diazotrophicus*, controlling siderophore uptake or other functions likely to be associated with iron metabolism, such as NADH-dependent electron transport and secondary metabolism. In addition, there is evidence that *fur* acts on *G. diazotrophicus* by coordinating the expression of genes involved in essential aspects of endophytic colonization and plant growth promotion. Our results are consistent with those of previous studies on *fur*-mediated regulation in *C. crescentus* [60], *A. ferrooxidans* [59], *N. gonorrhoeae* [61] and *Neisseria meningitidis* [62].

### 3.4. Molecular Characterization of fur from G. diazotrophicus

The generation of *fur* mutants was successfully performed in *E. coli*, *V. cholerae*, *Shigella flexneri*, *N. meningitidis* and *Burkholderia multivorans* [63,64,65,66,67]. However, in *P. aeruginosa*, *P. putida*, *Vibrio anguillarum*, *Neisseria gonorrhoeae*, *N. meningitidis*, *Nitrosomonas europaea* and some Gram-negative bacteria, mutation in the *fur* gene failed, as this event seems to be lethal [68,69,70,71]. The presence of only one copy of the *fur* gene in the *G. diazotrophicus* PAL5 genome may explain the essentiality of this gene for the viability of these bacteria in site-directed mutation events. 

Furthermore, the ability of the functional Fur protein of *G. diazotrophicus* to complement the *E. coli* H1780 *fur* mutant and the presence of *fur* box sequences in several genes reinforce the importance of Fur as a regulatory protein involved in a variety of cellular processes, including iron uptake and homeostasis, a response to oxidative stress and energy metabolism. Our results indicate that the *fur* gene encodes a functional Fur protein from *G. diazotrophicus* since it was able to complement *E. coli* H1780 cells, restoring the Fur phenotype of this strain. Similar results were found for *Nitrosomonas europaea* and *Bradyrhizobium japonicum,* where clones with a functional Fur protein were able to complement the *E. coli* H1780 *fur* mutant [71,72].

The accumulation of siderophores produced by *G. diazotrophicus* PAL5 in the LGI-P medium in the absence of iron was observed in *H. seropedicae,* where this limitation led to a reduction in the ability of the bacterium to transport the Fe^3+^-siderophore complex into the cell [73].

## 4. Materials and Methods

### 4.1. Bacterial Growth

*Gluconacetobacter diazotrophicus* PAL5 was obtained from the Diazotrophic Bacteria Culture Collection at the Biological Research Centre Johanna Döbereiner belonging to Embrapa Agrobiologia, Brazil. *G. diazotrophicus* wild type was grown in LGI-P liquid media [7]. 

The PAL5 strain was cultured in 5 mL of modified LGI-P medium at 30 °C and 180 rpm for approximately 48 h. After this period, a pre-inoculum (1.0 × 10^6^ cells) was suspended in 100 µL of saline solution [0.7% NaCl (*w*/*v*)] and inoculated in flasks containing 250 mL of modified LGI-P medium without the addition of FeCl_3_. The Erlenmeyer flask was then incubated at 30 °C and 180 rpm until the cell density reached the exponential growth phase (0.6 O.D._600_/mL) (Optic density at wavelength 600 nm). Subsequently, cultures were split into aliquots of 25 mL, put into 250 mL flasks and treated with FeCl_3_ (37 µM) or left untreated. This was done to ensure that all cells were in the same growth phase for transcriptome sequencing (RNA-seq) analysis. After 30 min of incubation, the cells were collected (5000 rpm, 10 min, 4 °C) and stored at −70 °C.

### 4.2. RNA-Seq Profiling Experiment

Total RNA was isolated with Trizol in accordance with the manufacturer’s protocol (Life Technologies, Carlsbad, CA, USA) and treated with DNase I (Epicentre, Madison, WI, USA) to remove genomic DNA contamination. RNA purity was quantified using Qubit (Life Technologies). Seven micrograms of total RNA were used for ribosomal RNA (rRNA) depletion with MICROBExpress^TM^ kit (Thermo Fisher Scientific Inc., Idstein, Germany). The efficiency of depletion was evaluated in agarose gel electrophoresis (1%), followed by quantification of the total RNA with an Agilent 2100 Bioanalyzer (Agilent Technologies, Santa Clara, CA, USA). A total of 500 ng mRNA was used for the construction of a sequencing library using the standard protocol of the SOLid Total RNAseq Kit (Life Technologies). The libraries were barcoded using the SOLiD Transcriptome Multiplexing Kit (Life Technologies). The emulsion PCR and sequencing were performed according to the Ion One Touch^TM^ 200 Template Kit v2 DL and the Ion PI^TM^ Sequencing 200 Kit v2 using the standard Life Technologies protocols, respectively. The Ion Proton Semiconductor Sequence (Life Technologies) was used to sequence 12 libraries generated from three biological replicates from each independent treatment.

### 4.3. Mapping, Clustering and Quality Control 

FASTQ files generated from sequencing were imported into CLC Bio’s Genomics Workbench v. 6.5 (Aarhus, Denmark). Sequences were then aligned with the genome sequence of *G. diazotrophicus* PAL5 (access number NC_010125.1) [15], via CLC Genomics Workbench version 7.5.1 program (Aarhus, Denmark). Reads containing ≤ 20 bp, and those mapping to rRNA 5S, 16S and 23S genes, were all discarded. The following parameters were used: the reads were trimmed for 20 bp, minimum 90% alignment with the reference sequence and 80% identity for inclusion as mapped read and the number of hits equal 1. 

The genes were considered expressed if more than three times the coverage was present. They were considered to be differentially expressed genes when the fold change was bigger than 1.2 or smaller than −1.2 and the *p*-value was smaller than 0.05 by *t*’ test. Visualizations were performed using Circos [74]. Gene ontology analysis was performed using the bioinformatics tool Blast2GO (https://www.blast2go.com/) (accessed on 10 May 2022) [75]. The genes were function classified by COG (Clusters if Orthologous Groups of proteins, (http://www.ncbi.nlm.nih.gov/COG) (accessed on 10 May 2022) [76].

### 4.4. Real-Time Quantitative (RT-qPCR) Validation

The isolated RNA sequencing samples were also used to perform a real-time quantitative (RT-qPCR) analysis. The cDNAs were synthesized using a Superscript™ III Reverse Transcriptase kit (Invitrogen, CA, USA). Gene expression was quantified using the GoTaq^®^ qPCR Master Mix (Promega, WI, USA) on a 7500 Fast Real-Time PCR System (Applied Biosystems, Foster City, CA, USA). Primer3 plus software [77] was used to design the primers. The *rhO* and *rpoD* genes were used as reference genes for RT-qPCR [78] and the relative gene expression was determined using qBase v.1.3.5 [79]. The Cq values from the genes evaluated in this study were obtained from Miner software (http://miner.ewindup.info/) (accessed on 10 May 2022). The PCR reaction consisted of 7.5 μL of GoTaq^®^ qPCR Master Mix. Different concentrations of forward and reverse primers (Appendix A) and 5.0 μL of 1:20 diluted cDNA template in a total volume of 15 μL were used. Cycling was performed using the default conditions of the 7500 Software v 2.0.5: 2 min at 95 °C, followed by 40 cycles of 20s at 95 °C and 30s at different temperatures. All RT-qPCR assays were carried out using three technical replication and non-template controls, as well as three independent cDNA syntheses. 

### 4.5. In Silico Searching for fur Binding Sites (fur Boxes)

The genome of *G. diazotrophicus* [15] was used for in silico *fur* Boxes searching with the Virtual Footprint online framework program [80]. The searches were carried out using the pre-existing *E. coli* (64-mer) and *P. aeruginosa* (16-mer) matrix [81]. The matrix was used to search the genome of *G. diazotrophicus* using a 19 bp sliding window. To further reduce the number of false positives, this initial pool of *fur* box candidates was culled by including only those that: (i) were located <600 nt from the proposed initiation of translation of the potential target gene and (ii) exhibited conservation of key nucleotides known to be protected by *fur* binding. Logos were generated using the web-based application available at http://weblogo.berkeley.edu/logo.cgi (accessed on 10 May 2022) [82].

### 4.6. Generation of fur Insertional Mutants of G. diazotrophicus

Two constructs carrying single Tn5 insertions at different positions inside the *fur* gene were selected and introduced into *G. diazotrophicus,* as described [18]. Putative transformants were selected in Dygs plates containing the selective antibiotics kanamycin (200 µg mL^−1^) and/or ampicillin (400 µg mL^−1^) [82,83].

### 4.7. Evaluation of the Production of Siderophores

The chromeazurol (CAS) assay was described by Schwyn and Neilands [84]. Briefly, 60.5 mg of CAS was dissolved in 50 mL of deionized water and mixed with 10 mL of a Fe^3+^ solution (1 mmol·L^−1^ FeCl_3_·6H_2_O, 10 mmol·L^−1^ HCl). While stirring, this solution was slowly mixed with 72.9 mg of hexadecyltrimethylammonium bromide (HDTMA), previously dissolved in 40 mL water. The resulting dark-blue solution was autoclaved, cooled to 50/60 °C, mixed with 900 mL sterile LGI-P and subsequently inoculated with bacterial strains and incubated in the dark (28 °C for 5 days).

### 4.8. Fur Titration Assays (FURTA)

Plasmids (Appendix A) were introduced into *E. coli* H1780 (*fur* inactivated) strains [64] and *lacZ* expression was assessed by visualization of a change in colony color from white to red on MacConkey lactose plates (BD Difco™, Rockville, MD, USA) supplemented with 30 μM ferrous ammonium sulfate. Plates were examined after 24 h of growth at 37 °C. The assays were performed in triplicate for each sample. Evaluation of the activity of the β-Galactosidase enzymes to *E. coli* H1780 strains, H1780 (pGGdfur) and H1780 (pGGd*fur*Tn5), was carried out after they were grown in two different conditions: medium minimum M9 liquid supplemented with iron and minimum M9 liquid medium with low iron availability due to the addition of 2 2′dipyridyl (2,2′-DIP).

## 5. Conclusions

In this study, the transcriptomic analysis provided an overview of the physiological strategies that *G. diazotrophicus* strain PAL5 employs for survival in iron-limiting conditions. Interestingly, the early transcriptional response of *G. diazotrophicus* goes beyond uptake and storage systems, encompassing T4SS, flagella, pili, chemotaxis, BNF and some transport proteins. This suggests that iron-sensing might be important in the early stages of plant colonization, in competition against other endophytes and in effective plant growth promotion. This characterized the *fur* transcriptional regulator and the identification of its regulon in the PAL5 strain, pointing to a model where *fur* controls the transcription of genes involved in iron homeostasis. The data in this study were useful for identifying genes involved in iron utilization in *G. diazotrophicus* and for shedding light on the adaptation mechanisms of iron-limited environments.

## Figures and Tables

**Figure 1 ijms-23-08533-f001:**
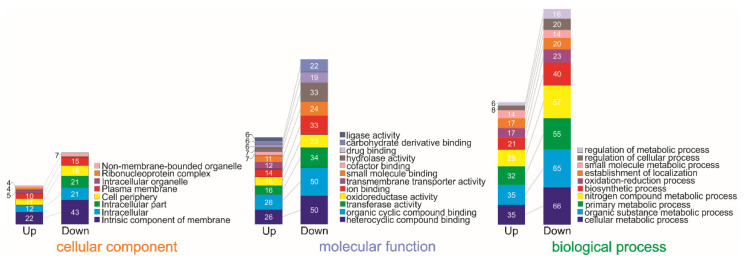
Functional classification of differentially expressed genes according to GO.

**Figure 2 ijms-23-08533-f002:**
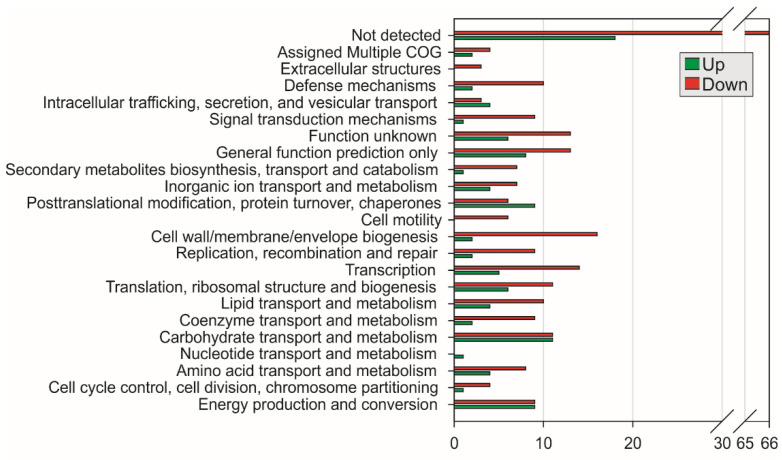
Functional classification of genes upregulated and downregulated in strain PAL5 grown under either iron or iron-depleted conditions. The genes were classified according to function by COG (Clusters of Orthologous Groups of proteins) (http://www.ncbi.nlm.nih.gov/COG (accessed on 10 May 2022)).

**Figure 3 ijms-23-08533-f003:**
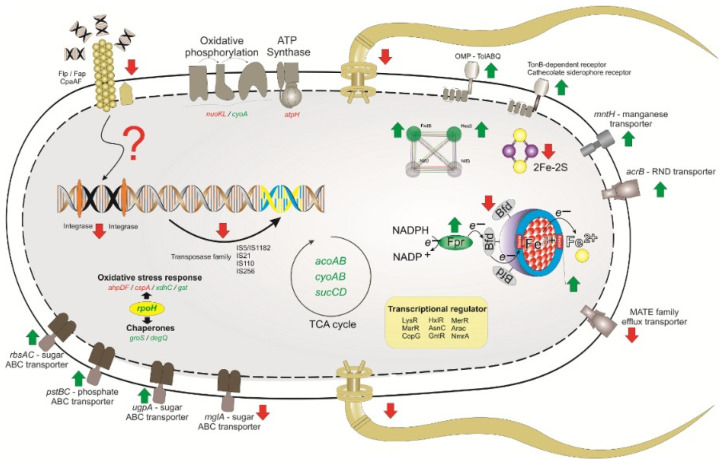
Overview of the major *Gluconacetobacter diazotrophicus* strain PAL5 metabolic pathways and functions differentially expressed in response to iron availability. Green arrows indicate general overexpression, while red arrows signal under expression.

**Figure 4 ijms-23-08533-f004:**
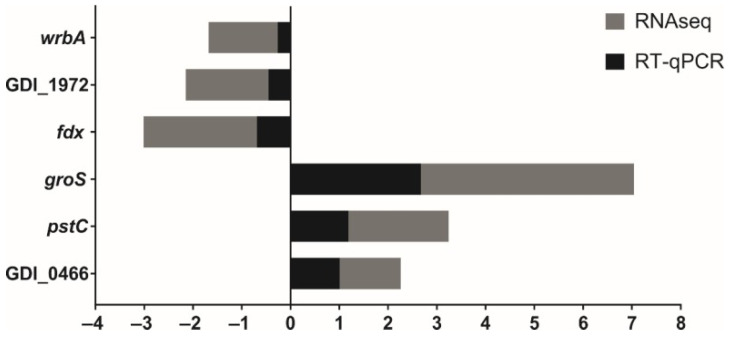
Expression pattern comparing RNA-seq and RT-qPCR analysis.

**Figure 5 ijms-23-08533-f005:**
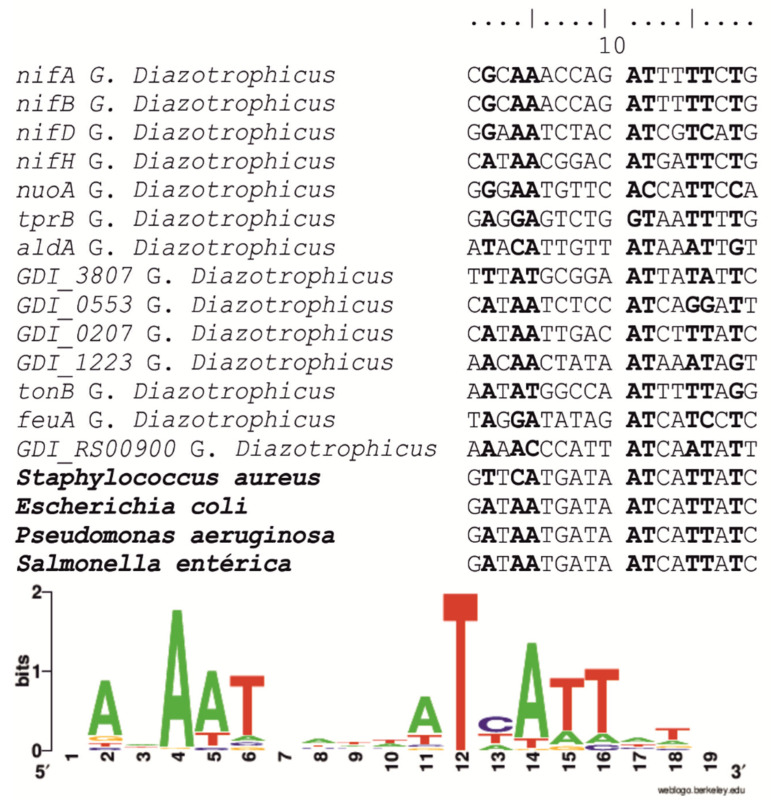
DNA sequence representing *G. diazotrophicus fur*-binding site. (Upper). Pairwise alignment of the derived *fur* box consensus sequences of *S. aureus*, *E. coli*, *P. aeruginosa S. enterica* and *G. diazotrophicus*. (Lower). Sequence logo of the *fur* box of *G. diazotrophicus* derived from bioinformatics and RNA-seq analyses. Base frequencies at each position were determined using WebLogo.

**Figure 6 ijms-23-08533-f006:**
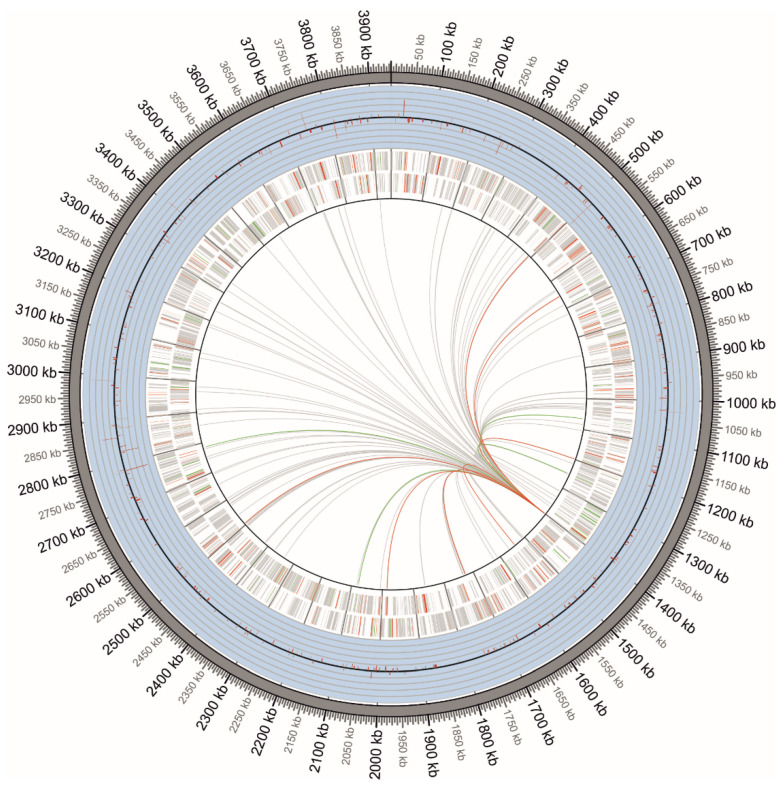
Circular representations of the *G. diazotrophicus* chromosome PAL5 showing transcriptome responses in the absence and presence of iron. On the chromosome, the outermost ring indicates the total transcriptome changes at the gene level, with the dark gray line representing the baseline and the light gray lines representing the fold-change changes. The two innermost rings indicate the genes in the forward and reverse chains. The green (up), red (down) and gray (non-expression) lines in the center indicate the regions likely to be regulated by the *fur*.

**Figure 7 ijms-23-08533-f007:**
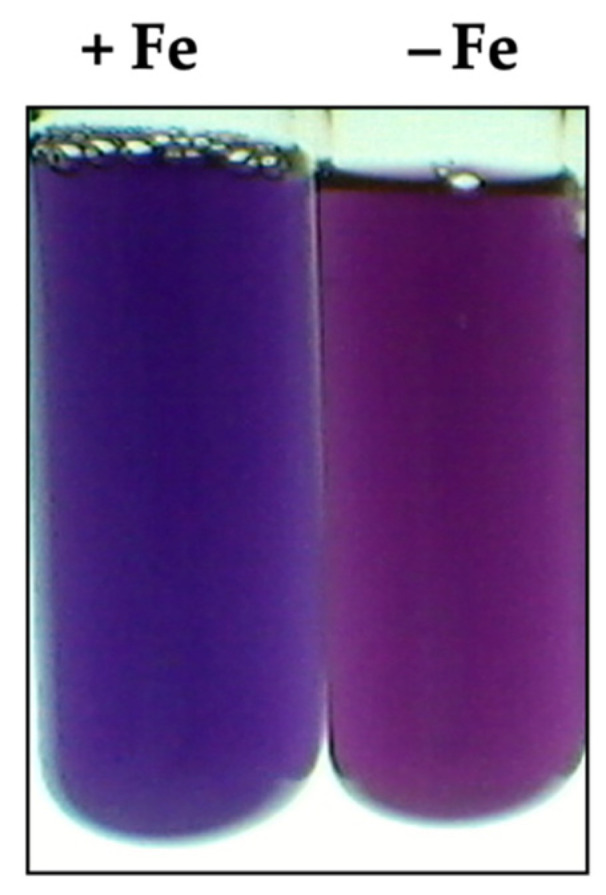
Production of siderophores by *G. diazotrophicus* PAL5 strains in response to iron availability.

**Figure 8 ijms-23-08533-f008:**
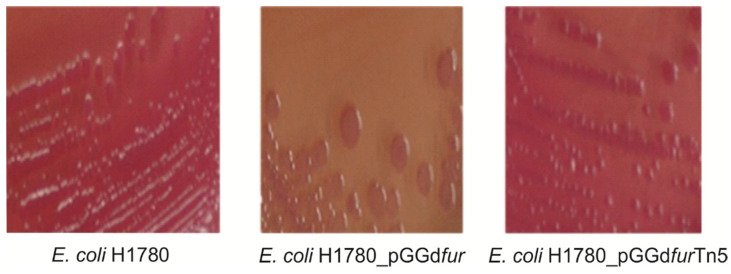
Fur Titration Assays (FURTA). Complementation of an *E. coli fur* mutant H1780 by *G. diazotrophicus fur* homologs. β-Galactosidase activity of *E. coli* H1780 cells containing the indicated plasmid plated in McConkey medium with 30 μM Fe supplement. All determinations are the means of at least three experiments, each in triplicate (Leica L2 stereoscope, magnification ×16).

**Figure 9 ijms-23-08533-f009:**
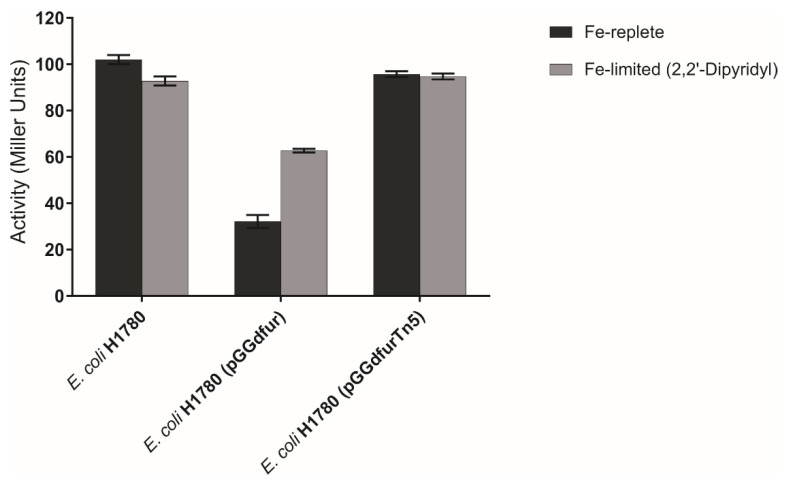
β-Galactosidase activity of *E. coli* H1780 cells containing the indicated plasmid and grown in M9 minimum medium liquid supplemented with iron and with low iron availability through the addition of 2,2′-DIP. All determinations are the means of at least three experiments (each in triplicate).

## Data Availability

RNA-Seq data have been deposited in the NCBI’s Gene Expression Omnibus (GEO) database [85] under the serial accession number GSE164445.

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
