# Peer review of "Transcriptomic Response of the Diazotrophic Bacteria Gluconacetobacter diazotrophicus Strain PAL5 to Iron Limitation and Characterization of the fur Regulatory Network"

_ijms, 2022, doi:10.3390/ijms23158533_

Round 1

Reviewer 1 Report

The submitted manuscript “Transcriptomic Response of the Diazotrophic Bacteria Gluconacetobacter diazotrophicus Strain PAL5 to Iron Limitation and Characterization of the fur Regulatory Network” by Soares et al., reports the early transcriptional response of Gram-negative bacterium G. diazotrophicus to iron. The data are nicely presented and could be helpful to understand the lifestyle and iron homeostasis of this bacterium. I think the following concerns need to be addressed prior to the publication.

1.     The introduction section needs considerable improvements. The Fur (ferric uptake regulator) proteins are well characterized in different bacterial species. The authors need to provide more information on the functions of Fur. Besides Fur, the author should include the previous reports on important iron regulators such as XibR, DtxR, and IdeR. Then authors need to establish connections with the present study and explain how this advances our understanding of iron homeostasis in bacteria.

2.     Lots of siderophore-mediated iron transport systems appeared in expression profiling. The simple CAS plate assays under different iron conditions, including those used for transcriptome analysis, could establish the importance of siderophores in this bacterium.

3.     The major claim about Fur remains unwarranted as the authors could not get fur mutant. Knocking out the fur is a difficult process as it is required to control the iron uptake system. Otherwise, the continuous accumulation of iron can induce cellular toxicity. Did the author try knocking out the Fur in presence of antioxidants? Sometimes creating point mutant or random mutagenesis or insertion at the c-terminal end of Fur has been helpful. Authors can also try to characterize the function of Fur by overexpressing.  

Author Response

Dear Reviewer 1,

The constructive criticisms of referees certainly contributed to the improvement of the manuscript. We attended most of the points raised by the reviewers, modified others and addressed detailed answers to those that we do not agree with them. Therefore, its quality has improved substantially and should be at a standard level of the IJMS.

The submitted manuscript “Transcriptomic Response of the Diazotrophic Bacteria Gluconacetobacter diazotrophicus Strain PAL5 to Iron Limitation and Characterization of the fur Regulatory Network” by Soares et al., reports the early transcriptional response of Gram-negative bacterium G. diazotrophicus to iron. The data are nicely presented and could be helpful to understand the lifestyle and iron homeostasis of this bacterium. I think the following concerns need to be addressed prior to the publication. --> We appreciate the compliments for the quality of the work and the interest in our results.

  1. The introduction section needs considerable improvements. The Fur (ferric uptake regulator) proteins are well characterized in different bacterial species. The authors need to provide more information on the functions of Fur. Besides Fur, the author should include the previous reports on important iron regulators such as XibR, DtxR, and IdeR. Then authors need to establish connections with the present study and explain how this advances our understanding of iron homeostasis in bacteria. --> We are grateful for the reviewer's contribution. We accept the suggestion by inserting paragraphs 5 and 6 in the introduction.
  1. Lots of siderophore-mediated iron transport systems appeared in expression profiling. The simple CAS plate assays under different iron conditions, including those used for transcriptome analysis, could establish the importance of siderophores in this bacterium. --> A CAS assays with siderophores were perform (Figure 7). The analysis improved the quality of the results and discussion.

3. The major claim about Fur remains unwarranted as the authors could not get fur mutant. Knocking out the fur is a difficult process as it is required to control the iron uptake system. Otherwise, the continuous accumulation of iron can induce cellular toxicity. Did the author try knocking out the Fur in presence of antioxidants? Sometimes creating point mutant or random mutagenesis or insertion at the c-terminal end of Fur has been helpful. Authors can also try to characterize the function of Fur by overexpressing.  --> We are grateful for the reviewer's concern regarding the existing iron concentrations in the culture medium, where excess iron can induce cytotoxicity, as well as the formation of reactive oxygen species. To avoid the effects of iron, we first grew all strains in culture medium containing dipyridyl (a strong iron chelating agent), thus mitigating the effects of iron. Second, we grew the strains in the presence of a minimal medium (with minimal iron concentrations), thus mitigating the effects of iron. Finally, we grew the strains in a culture medium, in which iron was replaced by manganese, thus mimicking the function of iron. Numerous random and site-directed knockout attempts were made (in different regions of the gene), but we were unable to succeed in generating the mutant. New results involving the over-expression of the Fur gene were generated and are being published in another journal. To clarify this point, we are presenting in this publication only the FURTA assay, where heterologous complementation was performed in a Fur-defective E. coli strain, where we managed to reverse the Fur-defective phenotype in E. coli from the Fur gene. from G. diazotrophicus.

Reviewer 2 Report

  The manuscript titled “Transcriptomic Response of the Diazotrophic Bacteria Gluconacetobacter diazotrophicus Strain PAL5 to Iron Limitation and Characterization of the fur Regulatory Network” submitted by Soares et al., describes the early transcriptional response of Gram-negative bacterium G. diazotrophicus to different levels of iron concentration. Presented data are well described and are impotrant for deeper understanding iron homeostasis of Gluconacetobacter diazotrophicus.
Below I have listed my remarks on the manuscript.

a) Since the Fur is well characterised in other bacterial species, the Authors could provide more info on this subject.

b) The importance of siderophores in this assay should be establish, by e.g. CAS plate with various iron concentrations.

c) The lack of fur mutant is troublesome, because it means the lack of experiemnt`s control. Perhaps a slight change of Fur knocking would help?. The other way could be overexpression of Fur gene, giving the chance to characterize its function.  

Author Response

Dear Reviewer 2,

The constructive criticisms of referees certainly contributed to the improvement of the manuscript. We attended most of the points raised by the reviewers, modified others and addressed detailed answers to those that we do not agree with them. Therefore, its quality has improved substantially and should be at a standard level of the IJMS.

The manuscript titled “Transcriptomic Response of the Diazotrophic Bacteria Gluconacetobacter diazotrophicus Strain PAL5 to Iron Limitation and Characterization of the fur Regulatory Network” submitted by Soares et al., describes the early transcriptional response of Gram-negative bacterium G. diazotrophicus to different levels of iron concentration. Presented data are well described and are impotrant for deeper understanding iron homeostasis of Gluconacetobacter diazotrophicus. Below I have listed my remarks on the manuscript. --> We appreciate the compliments for the quality of the work and the interest in our results.

a) Since the Fur is well characterised in other bacterial species, the Authors could provide more info on this subject. --> We are grateful for the reviewer's contribution. We accept the suggestion by inserting paragraphs 5 and 6 in the introduction.

b) The importance of siderophores in this assay should be establish, by e.g. CAS plate with various iron concentrations. --> A CAS assays with siderophores were perform (Figure 7). The analysis improved the quality of the results and discussion.

c) The lack of fur mutant is troublesome, because it means the lack of experiemnt`s control. Perhaps a slight change of Fur knocking would help?. The other way could be overexpression of Fur gene, giving the chance to characterize its function. --> We are grateful for the reviewer's concern regarding the existing iron concentrations in the culture medium, where excess iron can induce cytotoxicity, as well as the formation of reactive oxygen species. To avoid the effects of iron, we first grew all strains in culture medium containing dipyridyl (a strong iron chelating agent), thus mitigating the effects of iron. Second, we grew the strains in the presence of a minimal medium (with minimal iron concentrations), thus mitigating the effects of iron. Finally, we grew the strains in a culture medium, in which iron was replaced by manganese, thus mimicking the function of iron. Numerous random and site-directed knockout attempts were made (in different regions of the gene), but we were unable to succeed in generating the mutant. New results involving the over-expression of the Fur gene were generated and are being published in another journal. To clarify this point, we are presenting in this publication only the Fur Titration Assays (FURTA assay), where heterologous complementation was performed in a Fur-defective E. coli strain, where we managed to reverse the Fur-defective phenotype in E. coli from the Fur gene. from G. diazotrophicus.

Round 2

Reviewer 1 Report

The authors have addressed my concerns appropriately.

Author Response

Dear Reviewer 1,

We appreciate the compliments for the quality of the review and the interest in our results.

Best Regards

Reviewer 2 Report

Comments for the Authors:

The manuscript titled “Transcriptomic Response of the Diazotrophic Bacteria Gluconacetobacter diazotrophicus Strain PAL5 to Iron Limitation and Characterization of the fur Regulatory Network” submitted by Soares et al., describes the early transcriptional response of Gram-negative bacterium G. diazotrophicus to different levels of iron concentration. Presented data are well described and are important for a deeper understanding of iron homeostasis of Gluconacetobacter diazotrophicus.

In my opinion, the Authors have properly addressed their answers to all my remarks.

11)     The information about the Fur regulator has been extended and new data were added in the Introduction (lines 59-72)

22)     The importance of siderophores has been established using the CAS assay as suggested. The proper paragraphs were added in Results, and Materials & Method sections were added.

33)     Still, it would be nice to obtain the Fur mutant, however, the explanations provided by the Authors have convinced me that the effect of excess iron, which can induce cytotoxicity, as well as the formation of reactive oxygen species, has been taken into consideration and experimentally prevented by using culture medium containing dipyridyl (lines 245-248), and growing the strains in the presence of a minimal medium with minimal iron concentrations, mitigating the effects of iron, and by growing the strains on the medium in which iron was replaced by manganese. Although I could not find this information in the text. Perhaps, in the absence of Fur-mutant, it would be worth emphasising this information in the manuscript. This is my minor remark. Not obligatory. As compensation, the FUTRA assay was performed, where heterologous complementation was performed in a Fur-defective E. coli strain, where we the Authors have successfully reversed the Fur-defective phenotype in E. coli from the Fur gene. from G. diazotrophicus.

Author Response

Dear Reviewer 2,

The manuscript titled “Transcriptomic Response of the Diazotrophic Bacteria Gluconacetobacter diazotrophicus Strain PAL5 to Iron Limitation and Characterization of the fur Regulatory Network” submitted by Soares et al., describes the early transcriptional response of Gram-negative bacterium G. diazotrophicus to different levels of iron concentration. Presented data are well described and are important for a deeper understanding of iron homeostasis of Gluconacetobacter diazotrophicus. --> We appreciate the compliments for the quality of the work and the interest in our results.

In my opinion, the Authors have properly addressed their answers to all my remarks.

11)     The information about the Fur regulator has been extended and new data were added in the Introduction (lines 59-72). --> We appreciate the compliments for the quality of the work and the interest in our results.

22)     The importance of siderophores has been established using the CAS assay as suggested. The proper paragraphs were added in Results, and Materials & Method sections were added. --> We appreciate the compliments for the quality of the work and the interest in our results.

33)     Still, it would be nice to obtain the Fur mutant, however, the explanations provided by the Authors have convinced me that the effect of excess iron, which can induce cytotoxicity, as well as the formation of reactive oxygen species, has been taken into consideration and experimentally prevented by using culture medium containing dipyridyl (lines 245-248), and growing the strains in the presence of a minimal medium with minimal iron concentrations, mitigating the effects of iron, and by growing the strains on the medium in which iron was replaced by manganese. Although I could not find this information in the text. Perhaps, in the absence of Fur-mutant, it would be worth emphasising this information in the manuscript. This is my minor remark. Not obligatory. As compensation, the FUTRA assay was performed, where heterologous complementation was performed in a Fur-defective E. coli strain, where we the Authors have successfully reversed the Fur-defective phenotype in E. coli from the Fur gene. from G. diazotrophicus. --> We appreciate the compliments for the quality of the work and the interest in our results. As suggested by reviewer 2, we decided not to change the original text (reviewer 2 indicates that it is not mandatory to modify the text - "Although I could not find this information in the text. Perhaps, in the absence of Fur-mutant, it would be worth emphasising this information in the manuscript. This is my minor remark. Not obligatory).

Best Regards